# ProJo4D: Progressive Joint Optimization for Sparse-View Inverse Physics Estimation

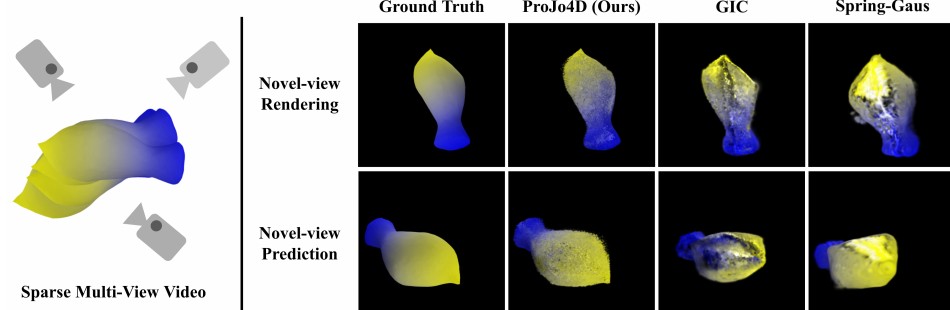

Figure 1: We present ProJo4D, a progressive joint optimization framework for estimating 4D representation and physical parameters of deformable objects from sparse multi-view video. ProJo4D significantly outperforms state-of-the-art inverse physics estimation algorithms, Spring-Gaus Zhong et al. (2024) and GIC Cai et al. (2024), which perform sequential optimization of scene geometry and physical parameters.

## Abstract

Neural rendering has advanced in 3D reconstruction and novel view synthesis. With the integration with physics, it opens up new applications. The inverse problem of estimating physics from visual data, however, remains challenging, limiting its effectiveness for applications like physically accurate digital twin creation in robotics and XR. Existing methods that incorporate physics into neural rendering frameworks typically require dense multi-view videos as input, making them impractical for scalable, real-world use. Given sparse multi-view videos, the sequential optimization strategy used by existing approaches introduces significant error accumulation, e.g., poor initial 3D reconstruction leads to inaccurate material parameter estimation in subsequent stages. Instead of sequential optimization, simultaneous optimization of all parameters also fails due to the highly non-convex and often non-differentiable nature of the problem. We propose ProJo4D, a progressive joint optimization framework that gradually increases the set of jointly optimized parameters, leading to fully joint optimization over geometry, appearance, physical state, and material property. Evaluations on both synthetic and real-world datasets show that ProJo4D outperforms prior work in 4D future state prediction and physical parameter estimation, demonstrating its effectiveness in physically grounded 4D scene understanding.

## 1 Introduction

Neural rendering techniques have made significant progress in 3D scene reconstruction and novel view synthesis Mildenhall et al. (2020); Müller et al. (2022); Kerbl et al. (2023), but they often lack adherence to the underlying physical laws (e.g., conservation of energy, momentum, or monotonicity constraints). This gap severely restricts their usage in downstream applications that require not only photorealistic appearance but also physically plausible behavior Chu et al. (2022). For instance, in vision-based robot learning Li et al. (2024); Jiang et al. (2025); Abou-Chakra et al. (2024), agents trained in simulations must seamlessly transfer the learned skills to the real world, which requires accurate physical interactions within the synthetic environment. Similarly, XR applications in many engineering and industrial settings require rendered objects to respond meaningfully to user interactions (e.g., changes in material properties or object dimensions), environmental constraints, and external forces to maintain immersion, usability, and seamless integration of virtual and physical worlds Jiang et al. (2024); Zheng et al. (2025).

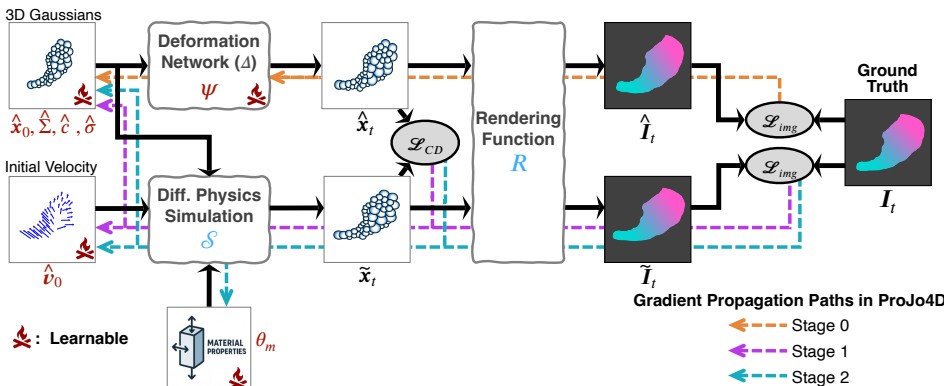

Figure 2: ProJo4D progressively grows the set of optimized variables—3D Gaussian parameters, deformation network, initial velocity, and material properties—across training stages to mitigate error propagation common in sequential frameworks like PAC-NeRF Li et al. (2023b) and GIC (Cai et al., 2024). The diagram illustrates the inter-dependencies among parameters in the inverse physics estimation task, with colored dotted arrows indicating gradient flow during each optimization stage.

A recent body of work Cai et al. (2024); Li et al. (2023b); Zhong et al. (2024); Chen et al. (2025) has attempted to bridge this gap by incorporating physics-based priors into neural rendering pipelines. However, the state-of-the-art approaches typically rely on dense multi-view setups, often requiring more than ten synchronized cameras with known poses. Such instrumentation imposes significant practical barriers, particularly in scenarios demanding scalable, flexible, or in-situ data collection. Whether in robot learning or industrial XR applications, the ability to create physically plausible digital twins from sparse observations is critical for real-world deployment. Overcoming the dependency on dense multi-view capture is thus crucial to realizing the full potential of neural rendering in physically grounded, deployable systems.

Sparse-view settings pose significant challenges for accurate 3D reconstruction and physical property estimation due to occlusions, shape ambiguities, and limited viewpoints. Existing methods that excel under dense observations degrade markedly when faced with sparse inputs, primarily due to the accumulation of errors in their sequential optimization pipelines Li et al. (2023b); Zhang et al. (2024); Huang et al. (2024); Zhong et al. (2024); Cai et al. (2024); Liu et al. (2025). These sequential optimization pipelines typically begin by learning an initial 3D or 4D scene representation from sparse images—often noisy and ambiguous—particularly in estimating geometry or particle positions. This flawed representation then serves as the basis for inferring physical state parameters (e.g., initial velocities) and subsequently material properties (e.g., stiffness, Poisson's ratio). As a result, errors introduced early propagate and compound, ultimately degrading both physical state and material parameter estimates. Although some recent works Zhong et al. (2024) have explored partial joint optimization of certain parameter subsets, they fall short of addressing the complete inverse physics problem from the outset. Fully joint optimization of all parameters remains challenging due to the highly non-convex, partly non-differentiable nature of the problem Zhong et al. (2021), often leading to poor local minima—particularly under sparse views.

To prove the effectiveness of progressive joint optimization strategy, without any model changes, we utilize GIC's 4D scene representation Cai et al. (2024) and physical models and only focus changing the optimization strategy. Through extensive evaluations, we demonstrate that our progressive joint optimization strategy significantly improves performance for inverse physics estimation on both synthetic and real-world datasets, mitigating the drastic performance drop in sparse-view scenarios. Our method outperforms the state-of-the-art GIC Cai et al. (2024), achieving superior results in 4D future state prediction (Chamfer Distance $16.11 \rightarrow 1.60$), rendering in the future prediction (PSNR $17.58 \rightarrow 22.30$), and physical parameter estimation (Poisson's ratio MAE $0.23 \rightarrow 0.10$, Young's Modulus MAE $0.18 \rightarrow 0.09$), as shown in Table 2.

## 2 BACKGROUND

### 2.1 NOTATION AND PROBLEM FORMULATION

We introduce the key notations and define the inverse physics estimation problem we aim to solve.

**Input Data.** We are given a set of input images $I = \{I_{t,c}\}_{t \in \mathcal{T}, c \in \mathcal{C}}$, where $I_{t,c}$ denotes an image captured at time $t$ from camera $c$. For each image $I_{t,c}$, the corresponding camera pose $P_c$ from a predefined set of cameras $c \in \mathcal{C}$ and the timestamp $t \in \mathcal{T}$ are assumed to be known. In addition a transparency alpha map $\alpha_{t,c}$ is often used for initial 3D/4D reconstruction, either rendered or estimated using segmentation or matting.

**3D / 4D Representation.** Our scene representation is based on 3D Gaussian Splatting (Kerbl et al., 2023). The 3D Gaussians are parameterized by their initial positions $\hat{\mathbf{x}}_0$, covariance matrices $\hat{\Sigma}$, color features $\hat{c}$, and opacity $\hat{\sigma}$. For representing 4D dynamics, we need additional parameters $\psi$. The position of a Gaussian at time $t$ is denoted by $\hat{\mathbf{x}}_t$ and is related to its initial position $\hat{\mathbf{x}}_0$ via a displacement function $\Delta(\cdot)$, which models the motion of the Gaussians over time, parameterized by $\psi$ as:

$$\hat{\mathbf{x}}_t = \hat{\mathbf{x}}_0 + \Delta(\hat{\mathbf{x}}_0, \psi, t). \tag{1}$$

We denote the rendering function, a differentiable splatting algorithm (Kerbl et al., 2023), by $R(\cdot, \cdot, \cdot, \cdot; \cdot)$. This function takes the Gaussian parameters ($\hat{\mathbf{x}}_t$, $\hat{\Sigma}$, $\hat{c}$, $\hat{\sigma}$) at time $t$ and the camera pose $P_c$ as input, and outputs a rendered image $\hat{I}$:

$$\hat{I}_{t,c} = R(\hat{\mathbf{x}}_t, \hat{\Sigma}, \hat{c}, \hat{\sigma}; P_c). \tag{2}$$

Similarly, $R_\alpha(\cdot, \cdot, \cdot; \cdot)$ denotes the alpha map rendering function and $\hat{\alpha}$ denotes the rendered alpha map. For detailed rendering process, please refer to 3D Gaussian Splatting (Kerbl et al., 2023).

**Physics Parameters.** Throughout the paper, we will refer to both the initial physical state $s$, such as initial velocity $v_0$, and material parameters $\theta_m$ as physical parameters. Material parameters $\theta_m$ include Young's modulus $E$ and Poisson's ratio $\nu$ for elastic objects. We will assume that the material model, e.g., elastic or plastic, is known a priori, consistent with all other prior works (Li et al., 2023b; Cai et al., 2024).

A physics simulation model, denoted by $\mathcal{S}(\cdot, \cdot, \cdot, \cdot)$, is used to predict the state of the system over time. Given the initial positions $\hat{\mathbf{x}}_0$, initial velocity $\hat{\mathbf{v}}_0$, material parameters $\hat{\theta}_m$, the simulation outputs the predicted positions $\tilde{\mathbf{x}}_t$ and its rendered image $\tilde{I}_{t,c}$ at time $t$ as:

$$\tilde{\mathbf{x}}_t = \mathcal{S}(\hat{\mathbf{x}}_0, \hat{\mathbf{v}}_0, \hat{\theta}_m, t), \quad \tilde{I}_{t,c} = R(\tilde{\mathbf{x}}_t, \hat{\Sigma}, \hat{c}, \hat{\sigma}; P_c). \tag{3}$$

**Problem Formulation.** In summary, our task is an inverse estimation problem: given input images $I$, their corresponding camera parameters $P$, we aim to estimate the underlying geometry $\mathbf{x}_0$, appearance parameters $(\Sigma, c, \sigma)$, and physical properties $(v_0, \theta_m)$ of a deformable object.

## 2.2 Related Works

**Differentiable Physics Simulation.** Differentiable physics simulation is being widely used to optimize and estimate physics-related parameters (Xu et al., 2019; Sanchez-Gonzalez et al., 2020; Hu et al., 2020; Geilinger et al., 2020; Zhong et al., 2021; Murthy et al., 2021; Wang et al., 2024). This forms the foundation of this research area, raising important considerations about which differentiable simulation frameworks to employ and how to design and schedule the optimization process. Among the commonly used simulation methods are the spring-mass model (Zhong et al., 2024) and the Material Point Method (MPM) (Jiang et al., 2016). The gradients from the simulations enable updating physical parameters for system identification. Simplicits (Modi et al., 2024) can be used for accelerated inverse physics (Chen et al., 2025), but it only supports (hyper)elastic materials with simple gravity-only scenarios. We use differentiable MPM as the physics simulator $\mathcal{S}(\cdot, \cdot, \cdot, \cdot)$.

**Physics-based Neural Rendering.** Existing neural rendering techniques that integrate physics can be broadly categorized into two main directions: accurately estimating physical parameters from videos (Li et al., 2023a; Yu et al., 2023; Kaneko, 2024; Xue et al., 2023; Qiao et al., 2022; Ma et al., 2021; Guan et al., 2022), and generating plausible parameters or realistic dynamics from a static scene (Zhai et al., 2024; Liu et al., 2024b;a).

In one of the early attempts in accurately estimating physical parameters, PAC-NeRF (Li et al., 2023b), proposed a general framework, optimizing geometry and appearance, initial physical state, and materials sequentially. Spring-Gaus (Zhong et al., 2024) introduced a spring-mass model applied to 3D Gaussian Splatting (Kerbl et al., 2023). Gaussian Informed Continuum (GIC) (Cai

et al., 2024) further improves the physical parameter estimation and future prediction by using learned 4D representations during physical parameter estimation to provide guidance for 3D losses. Vid2Sim (Chen et al., 2025) further proposed using pretrained models for initial material parameter estimates before per-scene optimization and using Simplicits Modi et al. (2024) for accelerated optimization.

Methods in the generative category often synthesize realistic videos using video diffusion models Zhang et al. (2024); Huang et al. (2024); Lin et al. (2025). Using the generated videos and differentiable physics simulation, they estimate plausible parameters. Some approaches also incorporate LLMs to get initial parameters Liu et al. (2024b; 2025). While successful at producing visually compelling dynamics, the primary objective here is typically visual plausibility and motion generation rather than the accurate estimation of physical parameters from target observations.

Our method contributes to the first category by addressing accurate physical parameter estimation from videos, specifically under sparse views. While existing inverse physics estimation techniques perform sequential optimization Li et al. (2023b); Zhong et al. (2024); Cai et al. (2024), as shown in Tab. 1, leading to error propagation across multiple stages, ProJo4D introduces a progressive joint optimization strategy that yield large improvements in sparse-view scenarios.

Table 1: Optimization strategies of existing methods. X, A, S, and M denote positions, appearances, physical states, and material parameters. 0 denotes initial 3D/4D representation learning before physical parameter estimation. △ denotes optional optimization, depending on the scene.

| Method | Param. | 0 | 1 | 2 | 3 | 4 |
|---|---|---|---|---|---|---|
| | | | | Stage | | |
| PAC-NeRF Li et al. (2023b) PhysDreamer Zhang et al. (2024) | X | ✓ | | | | |
| | A | ✓ | | | | |
| | S | | ✓ | | | |
| | M | | | ✓ | | |
| Spring-Gaus Zhong et al. (2024) | X | ✓ | | | | |
| | A | ✓ | △ | | △ | △ |
| | S | | | ✓ | | |
| | M | | | | | ✓ |
| GIC Cai et al. (2024) MASIV Zhao et al. (2025) | X | ✓ | | | | |
| | A | ✓ | | | ✓ | |
| | S | | ✓ | | | |
| | M | | | ✓ | | |
| Vid2Sim Chen et al. (2025) | X | ✓ | ✓ | | | |
| | A | ✓ | ✓ | | | |
| | S | | | | | |
| | M | ✓ | ✓ | | | |
| Ours | X | ✓ | ✓ | ✓ | | |
| | A | ✓ | ✓ | ✓ | | |
| | S | | ✓ | ✓ | | |
| | M | | | ✓ | | |

# 3 PROJO4D

For a given scene, our approach follows a multi-stage pipeline to estimate the constituting parameters: the appearance of a deformable object, initial physical states, and material properties. As is common in this domain, our pipeline begins with obtaining an initial 3D/4D representation. Our primary focus lies in the subsequent progressive joint optimization strategy designed to robustly solve inverse physics estimation from limited observations.

## 3.1 MOTIVATION

**Physical Parameter Estimation.** Estimating the physical state and material parameters of an object from visual observations is a challenging inverse problem. This is difficult primarily because the whole system is non-linear and non-convex. To make it worse, some material models, like non-Newtonian fluids, have non-differentiable but material-parameter-dependent branches. In addition, some of the physical parameters are strongly coupled, making it difficult to disambiguate their individual contributions from visual cues. All these difficulties necessitate careful design of optimization strategies to improve the chances of converging to a physically plausible and accurate solution.

**Error accumulation in sparse vs. dense views under sequential optimization.** As discussed above, most existing methods rely on sequential optimization pipelines Li et al. (2023b); Cai et al. (2024); Zhang et al. (2024), where parameters are optimized in stages and estimates from earlier stages are fixed as inputs for later ones. While this strategy can help mitigate some challenges, it introduces a new problem: errors from earlier stages propagate and accumulate, with the effect being substantially worse for sparse-view settings compared to dense ones.

For sparse views, the initial geometry estimation is considerably less accurate, and these errors cas-

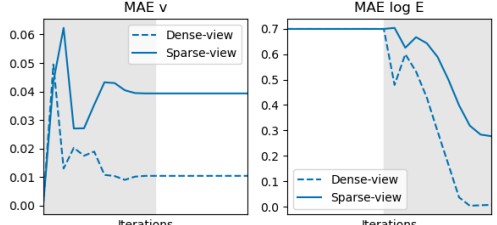

Figure 3: Comparison of error propagation in sequential optimization. The gray region marks the iterations during which the corresponding parameter is optimized: velocity (MAE-$v$, left) and material stiffness (MAE-$\log E$, right). Dense views reduce errors faster; sparse views accumulate more errors.

cade through subsequent optimization stages. This leads to large errors in estimating both initial states and material properties. Figure 3 illustrates this phenomenon by plotting how mean absolute errors (MAE) in velocity ($v$) and material parameters ($\log E$) evolve during the sequential optimization. We exclude the shared initial stage, where the 3D/4D representation is constructed, and focus on the following two stages: velocity optimization (MAE $v$; left) and material parameter optimization (MAE $\log E$; right). With dense views, errors in the initial 3D/4D representation are smaller, leading to reduced error propagation in subsequent stages, whereas sparse views suffer from greater error accumulation across stages. For real-world deployments, capturing dense multi-view data with precisely calibrated and synchronized cameras is often impractical. Consequently, mitigating error accumulation and propagation becomes essential to extend the applicability of physics-based 4D reconstruction methods in real-world scenarios.

**Choice of optimization strategies: sequential vs. joint vs. progressive.** Figure 4 illustrates how optimization strategies affect estimation accuracy and robustness across different object shapes and material models. We plot the error in material properties ($\log E$ for elastic (a) and $\theta_\alpha$ for sand (b)) and in future 4D state simulation (EMD) over optimization iterations. Sequential optimization, while common, suffers from significant error accumulation as shown in Fig. 4a, leading to high estimation errors.

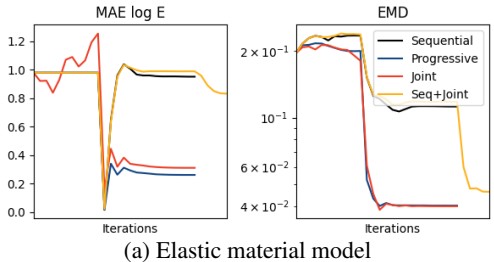

(a) Elastic material model

An alternative is joint optimization, adopted by recent approaches such as Vid2Sim Chen et al. (2025). This strategy can be effective for relatively simple models like elastic objects (Fig. 4a), but struggles with more complex systems such as sand (Fig. 4b) or non-Newtonian materials, where the optimization landscape is highly non-convex. In such cases, while initial geometry may already be close to ground truth, the physical parameters

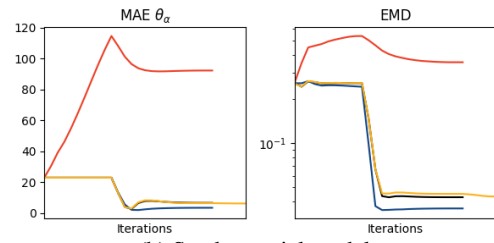

(b) Sand material model

Figure 4: Material parameter estimation and future prediction performances of different optimization strategies in different material models.

are typically far from accurate, causing optimization to stagnate in sub-optimal regions. A hybrid variant—performing joint optimization after sequential optimization—attempts to address this but still inherits the limitations of the initial sequential stage. For example, when sequential optimization fails for elastic objects, subsequent joint refinement yields only limited improvements compared to alternatives (Fig. 4a).

These findings highlight the need for more principled optimization strategies that generalize across diverse object shapes and material models. In the following section, we introduce our progressive joint optimization approach, which mitigates these issues and consistently reduces estimation errors, improving both performance and robustness.

## 3.2 PROGRESSIVE JOINT OPTIMIZATION

**Stage 0: Initial 3D / 4D Representation Learning.** Most existing pipelines, including ours, start with an initial 3D or 4D representation learning stage. Some methods Li et al. (2023b); Zhong et al. (2024) learns a static 3D scene from the first image of each camera, while others Cai et al. (2024) learn a full 4D representation from the multi-view image sequence. Our method is based on 4D representation learning to leverage 3D guidance during the following stages. The parameters for 4D representation are optimized by minimizing rendering losses over all frames and cameras:

$$\mathcal{L}_{img}(\hat{I}, I) = \lambda_{L1}\mathcal{L}_{L1}(\hat{I}, I) + \lambda_{SSIM}\mathcal{L}_{SSIM}(\hat{I}, I), \tag{4}$$

$$\hat{\mathbf{x}}_0^*, \hat{\Sigma}^*, \hat{c}^*, \hat{\sigma}^*, \psi^* = \underset{\hat{\mathbf{x}}_0, \hat{\Sigma}, \hat{c}, \hat{\sigma}, \psi}{\arg\min} \sum_{t \in \mathcal{T}} \sum_{c \in \mathcal{C}} \lambda_{img}\mathcal{L}_{img}\left(\hat{I}_{t,c}, I_{t,c}\right) + \lambda_\alpha\mathcal{L}_{L1}\left(\hat{\alpha}_{t,c}, \alpha_{t,c}\right), \tag{5}$$

where $\hat{I}_{t,c}$ and $\hat{\alpha}_{t,c}$ denote a rendered image and an alpha map, respectively (Sec. 2.1). $\mathcal{L}_{L1}$ and $\mathcal{L}_{SSIM}$ denote L1 loss and Structural similarity index measure (SSIM) loss, and $\lambda_{L1}, \lambda_{SSIM}$ are their corresponding loss weights.

**Stage 1: Initial Physical State Optimization.** This is the first stage of our progressive optimization strategy, focusing on estimating initial physical state parameters $s$, such as the initial velocity $\hat{v}_0$. In this stage, we use the first few frames, following prior works Li et al. (2023b); Zhong et al. (2024); Cai et al. (2024). This allows the optimization to focus on estimating initial velocity $\hat{v}_0$ before significant deformation or complex interactions take place, separating the influence of initial motion from material response. At this stage, Gaussian parameters are also optimized. As the initial velocity $v_0$ is the only physical state parameters in most existing benchmarks, we only optimize $\hat{v}_0$ by minimizing a combined loss over the first few frames $\mathcal{T}_k$:

$$\hat{v}_0^*, \hat{\mathbf{x}}_0^*, \hat{\Sigma}^*, \hat{c}^*, \hat{\sigma}^* = \underset{\hat{v}_0, \hat{\mathbf{x}}_0, \hat{\Sigma}, \hat{c}, \hat{\sigma}}{\arg\min} \quad \lambda_{img} \sum_{t \in \mathcal{T}} \sum_{c \in \mathcal{C}} \mathcal{L}_{img}\Big(\tilde{I}_{t,c}, I_{t,c}\Big) + \lambda_{geo} \sum_{t \in \mathcal{T}} \mathcal{L}_{geo}\Big(\tilde{\mathbf{x}}_t, \hat{\mathbf{x}}_t\Big), \quad (6)$$

where $\mathcal{L}_{geo}$ is the bidirectional chamfer distance, which measures the distance to the closest point from both estimated and ground truth points. $\tilde{\mathbf{x}}_t$ and $\tilde{I}_{t,c}$ are positions from physics simulation $\mathcal{S}(\cdot)$ and their corresponding rendered image (Sec. 2.1). $\hat{\mathbf{x}}_t$ denotes the extracted positions from the learned 4D representation using deformation network $\Delta(\cdot)$, as proposed in GIC Cai et al. (2024).

**Stage 2: Full Joint Optimization.** After obtaining an improved estimate for physical states $s$, more specifically $v_0$, this stage progresses to include material parameters $\hat{\theta}_m$ during optimization. For this stage, we utilize data from all frames. We use the same optimization objective as the previous stage:

$$\hat{v}_0^*, \hat{\theta}_m^*, \hat{\mathbf{x}}_0^*, \hat{\Sigma}^*, \hat{c}^*, \hat{\sigma}^* = \underset{\hat{v}_0, \hat{\theta}_m, \hat{\mathbf{x}}_0, \hat{\Sigma}, \hat{c}, \hat{\sigma}}{\arg\min} \quad \lambda_{img} \sum_{t \in \mathcal{T}} \sum_{c \in \mathcal{C}} \mathcal{L}_{img}\Big(\tilde{I}_{t,c}, I_{t,c}\Big) + \lambda_{geo} \sum_{t \in \mathcal{T}} \mathcal{L}_{geo}\Big(\tilde{\mathbf{x}}_t, \hat{\mathbf{x}}_t\Big).$$
$$(7)$$

Table 2: Evaluation on Spring-Gaus Zhong et al. (2024) dataset with 7 elastic objects of different shapes and physical parameters, captured with sparse views (3 views). We measure 3D prediction accuracy of future states using Chamfer Distance (CD) and Earth Movers Distance (EMD), image rendering quality of future states using PSNR and SSIM, and MAE of Young's Modulus ($E$) and Poisson Ratio ($\nu$). For rendering quality evaluation, we used future images from all cameras.

|  | method | apple | banana | chess | cream | cross | paste | torus | mean |
|---|---|---|---|---|---|---|---|---|---|
| CD ↓ | Spring-Gaus | 12.12 | 51.35 | 3.68 | 2.97 | 40.30 | 73.08 | 15.00 | 26.93 |
|  | GIC | 2.13 | 8.37 | 7.51 | 8.16 | 2.51 | 81.24 | 2.81 | 16.11 |
|  | GIC + ProJo4D | **0.19** | **0.12** | **1.37** | **1.54** | **0.38** | **6.93** | **0.65** | **1.60** |
| EMD ↓ | Spring-Gaus | 0.170 | 0.223 | 0.097 | 0.101 | 0.232 | 0.248 | 0.177 | 0.178 |
|  | GIC | 0.090 | 0.106 | 0.139 | 0.135 | 0.084 | 0.263 | 0.081 | 0.128 |
|  | GIC + ProJo4D | **0.054** | **0.024** | **0.066** | **0.052** | **0.031** | **0.142** | **0.031** | **0.057** |
| PSNR ↑ | Spring-Gaus | 17.03 | 15.79 | 13.85 | 14.62 | 11.24 | 10.94 | 13.01 | 13.78 |
|  | GIC | 20.52 | 21.84 | 14.87 | 13.93 | 22.51 | 12.41 | 17.00 | 17.58 |
|  | GIC + ProJo4D | **27.10** | **28.65** | **17.96** | **18.76** | **28.09** | **15.20** | **20.354** | **22.30** |
| SSIM ↑ | Spring-Gaus | 0.790 | 0.825 | 0.792 | 0.796 | 0.819 | 0.737 | 0.831 | 0.799 |
|  | GIC | 0.868 | 0.910 | 0.826 | 0.795 | 0.889 | 0.772 | 0.892 | 0.850 |
|  | GIC + ProJo4D | **0.930** | **0.959** | **0.886** | **0.885** | **0.943** | **0.852** | **0.933** | **0.913** |
| MAE $\log E$ ↓ | GIC | 0.1840 | 0.4639 | 0.1807 | 0.0838 | 0.3239 | **0.1380** | 0.2436 | 0.2311 |
|  | GIC + ProJo4D | **0.0633** | **0.1519** | **0.0326** | **0.0336** | **0.0469** | 0.1705 | **0.2315** | **0.1043** |
| MAE $\nu$ ↓ | GIC | 0.1439 | **0.1049** | 0.0622 | 0.1407 | 0.0955 | 0.2209 | 0.4851 | 0.1790 |
|  | GIC + ProJo4D | **0.0817** | 0.2237 | **0.0222** | **0.0295** | **0.0307** | **0.0928** | **0.1569** | **0.0911** |

# 4 EXPERIMENTS

## 4.1 EXPERIMENTAL SETTINGS

**Baselines.** We compared ours with the current state-of-the-art methods; PAC-NeRF Li et al. (2023b), Spring-Gaus Zhong et al. (2024), GIC Cai et al. (2024), and Vid2Sim Chen et al. (2025).

**Datasets.** We used the synthetic dataset from Spring-Gaus Zhong et al. (2024), PAC-NeRF Li et al. (2023b), and the GSO dataset from Vid2Sim Chen et al. (2025). In the Spring-Gaus dataset, 7 distinct object shapes are provided, whereas the GSO dataset offers 12 shapes; both datasets only

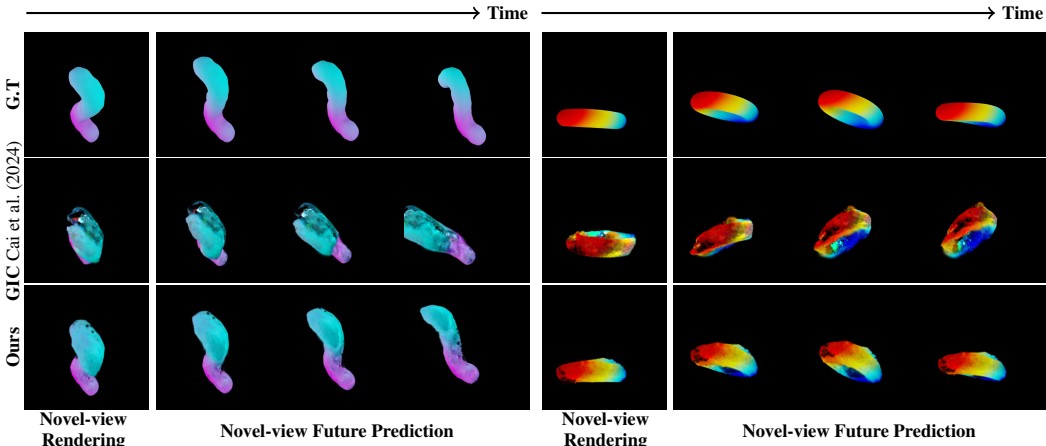

Figure 5: Visual comparison of ProJo4D(Ours) with GIC Cai et al. (2024) for novel-view rendering and prediction in future timestep on the Spring-Gaus dataset, using sparse-view inputs. ProJo4D produces more consistent and physically plausible results across both current and future views.

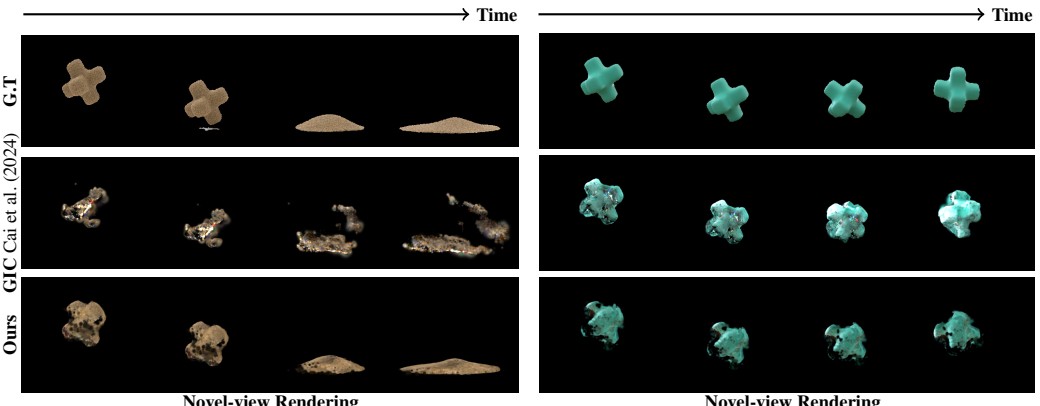

Figure 6: Visual comparison of ProJo4D(Ours) with GIC Cai et al. (2024) for novel-view rendering on 'Sand' (left) and 'elastic' (right) materials from the PAC-NeRF dataset, using sparse-view inputs. ProJo4D reconstructs significantly better geometry than GIC.

involve elastic materials with varying parameters. We used the PAC-NeRF synthetic dataset that comprises five different material models: elastic (Neo-Hookean), Newtonian fluid, non-Newtonian fluid, plasticine, and sand (Drucker-Prager), each sharing the same object shape but with different physical parameters. The dataset has a total of 45 scenes, and we report the mean and standard deviation for each material model. To evaluate how our proposed optimization strategy can improve performance in sparse-view settings, we selected only three cameras from the ten cameras available for both datasets: the second, sixth, and tenth cameras from each scene using Spring-Gaus and PAC-NeRF datasets. For the GSO dataset, we used the same experimental settings as Vid2Sim. To compare with other baselines, we used the same train/test splits provided by each dataset.

For real-world evaluation, we use the Spring-Gaus dataset, which provides exactly three cameras for each scene. Following Spring-Gaus Zhong et al. (2024), we optimize with the same data preprocessing steps. We train each model on the first 14 frames and test on the remaining 6 frames.

**Metrics.** To evaluate our method against state-of-the-art approaches, we adopt two categories of metrics: future state prediction and physical parameter estimation, following prior works Li et al. (2023b); Zhong et al. (2024); Cai et al. (2024). For future state prediction, we measure the 3D discrepancy between simulated positions $\tilde{x}_t$ and ground-truth positions $x_t$ using Chamfer Distance (CD) and Earth Mover's Distance (EMD). We also assess the 2D rendering quality of predicted future states from both seen viewpoints (three in our experiments) and novel viewpoints using peak signal-to-noise ratio (PSNR) and structural similarity index measure (SSIM). For physical parameter

Table 3: Evaluation on PAC-NeRF dataset Li et al. (2023b), containing five material types with ten different parameter settings each (except sand, which has five), all sharing the same object shape. We report 3D future state prediction accuracy using Chamfer Distance (CD) and Earth Mover's Distance (EMD), and evaluate material estimation with mean absolute error (MAE) across parameters. "N/A" indicates parameters not applicable to a given material type.

| | | Elasticity | Newtonian | Non-Newtonian | Plasticine | Sand |
|---|---|---|---|---|---|---|
| CD ↓ | GIC | $5.512 \pm 3.311$ | $0.537 \pm 0.315$ | $0.689 \pm 0.398$ | $2.012 \pm 1.797$ | $20.262 \pm 43.360$ |
| | + ProJo4D | $\mathbf{0.913 \pm 0.301}$ | $\mathbf{0.339 \pm 0.108}$ | $\mathbf{0.473 \pm 0.248}$ | $\mathbf{1.103 \pm 0.948}$ | $\mathbf{0.264 \pm 0.017}$ |
| | + Full Joint | $1.318 \pm 1.117$ | $0.346 \pm 0.095$ | $8.104 \pm 13.563$ | $17.678 \pm 18.170$ | $53.564 \pm 19.404$ |
| EMD ↓ | GIC | $0.126 \pm 0.041$ | $0.103 \pm 0.007$ | $0.040 \pm 0.007$ | $0.062 \pm 0.027$ | $0.122 \pm 0.162$ |
| | + ProJo4D | $\mathbf{0.042 \pm 0.007}$ | $\mathbf{0.039 \pm 0.004}$ | $\mathbf{0.038 \pm 0.005}$ | $\mathbf{0.053 \pm 0.018}$ | $\mathbf{0.045 \pm 0.006}$ |
| | + Full Joint | $0.049 \pm 0.023$ | $0.040 \pm 0.005$ | $0.099 \pm 0.073$ | $0.124 \pm 0.074$ | $0.223 \pm 0.020$ |
| MAE $v_0$ ↓ | GIC | $0.008 \pm 0.004$ | $0.009 \pm 0.004$ | $0.015 \pm 0.008$ | $\mathbf{0.010 \pm 0.005}$ | $0.007 \pm 0.004$ |
| | + ProJo4D | $\mathbf{0.007 \pm 0.003}$ | $\mathbf{0.008 \pm 0.002}$ | $\mathbf{0.005 \pm 0.003}$ | $0.024 \pm 0.056$ | $\mathbf{0.005 \pm 0.003}$ |
| | + Full Joint | $0.020 \pm 0.033$ | $0.008 \pm 0.004$ | $0.080 \pm 0.099$ | $0.092 \pm 0.102$ | $0.046 \pm 0.032$ |
| MAE $\log(E)$ ↓ | GIC | $0.189 \pm 0.217$ | N/A | N/A | $1.597 \pm 1.150$ | N/A |
| | + ProJo4D | $\mathbf{0.124 \pm 0.099}$ | N/A | N/A | $\mathbf{0.742 \pm 0.780}$ | N/A |
| | + Full Joint | $0.216 \pm 0.299$ | N/A | N/A | $2.856 \pm 2.196$ | N/A |
| MAE $\nu$ ↓ | GIC | $0.123 \pm 0.103$ | N/A | N/A | $0.134 \pm 0.112$ | N/A |
| | + ProJo4D | $\mathbf{0.048 \pm 0.034}$ | N/A | N/A | $\mathbf{0.084 \pm 0.029}$ | N/A |
| | + Full Joint | $0.061 \pm 0.053$ | N/A | N/A | $0.075 \pm 0.057$ | N/A |
| MAE $\log(\mu)$ ↓ | GIC | N/A | $\mathbf{0.103 \pm 0.125}$ | $0.869 \pm 0.598$ | N/A | N/A |
| | + ProJo4D | N/A | $0.134 \pm 0.175$ | $\mathbf{0.491 \pm 0.363}$ | N/A | N/A |
| | + Full Joint | N/A | $0.294 \pm 0.314$ | $2.315 \pm 1.100$ | N/A | N/A |
| MAE $\log(\kappa)$ ↓ | GIC | N/A | $3.180 \pm 1.085$ | $0.725 \pm 0.704$ | N/A | N/A |
| | + ProJo4D | N/A | $\mathbf{1.425 \pm 1.148}$ | $\mathbf{0.462 \pm 0.344}$ | N/A | N/A |
| | + Full Joint | N/A | $3.312 \pm 1.679$ | $1.673 \pm 1.918$ | N/A | N/A |
| MAE $\log(\tau_Y)$ ↓ | GIC | N/A | N/A | $\mathbf{0.069 \pm 0.069}$ | $0.327 \pm 0.365$ | N/A |
| | + ProJo4D | N/A | N/A | $0.144 \pm 0.071$ | $\mathbf{0.144 \pm 0.125}$ | N/A |
| | + Full Joint | N/A | N/A | $1.839 \pm 3.242$ | $6.612 \pm 7.739$ | N/A |
| MAE $\log(\eta)$ ↓ | GIC | N/A | N/A | $0.519 \pm 0.264$ | N/A | N/A |
| | + ProJo4D | N/A | N/A | $\mathbf{0.463 \pm 0.244}$ | N/A | N/A |
| | + Full Joint | N/A | N/A | $1.455 \pm 2.263$ | N/A | N/A |
| MAE $\theta_{fric}$ ↓ | GIC | N/A | N/A | N/A | N/A | $6.785 \pm 8.458$ |
| | + ProJo4D | N/A | N/A | N/A | N/A | $\mathbf{4.998 \pm 2.542}$ |
| | + Full Joint | N/A | N/A | N/A | N/A | $67.893 \pm 12.416$ |

estimation, we report mean absolute error (MAE), following previous literature Li et al. (2023b); Zhong et al. (2024); Cai et al. (2024).

**Hypeparameters.** To focus on the optimization strategy, we use the same learning rates as GIC for both Spring-Gaus and the PAC-NeRF datasets. We also optimized with the same number of iterations for Stage 1 and 2 as GIC. We set the loss weights for image loss $\lambda_{img}$ and Chamfer distance $\lambda_{geo}$ to $1/|\mathcal{C}|$ and 1.0, respectively (Eqs. 6 and 7).

## 4.2 RESULTS

**Synthetic Datasets.** The *Spring-Gaus dataset* evaluates performance across diverse object shapes and appearances. As shown in Tab. 2 and Fig. 7, both Spring-Gaus and GIC degrade significantly under sparse views, especially on trajectory-sensitive scenes such as toothpaste, whereas our method remains considerably more robust. Fig. 5 further illustrates that existing approaches fail to estimate geometry and physical parameters, critically failing to predict future trajectories. In contrast, our method remains robust with sparse views, achieving a tenfold reduction in Chamfer Distance, roughly half the Earth Mover's Distance on average, and a substantial PSNR boost ($17.58 \rightarrow 22.30$).

The *PAC-NeRF dataset* evaluate the accuracy and robustness to different material models and their parameters. Table. 3 and Fig. 6 show both quantitatively and qualitatively that our methods outperforms GIC across different material models and different metrics and parameters. Both non-Newtonian and Plasticine, has non-differentiable branch, which poses additional challenge to the progressive joint optimization strategy. Nevertheless, our method improves upon GIC in 4/5 paramaters for Non-Newtonian and in 3/4 parameters for Plasticine, and strong improvement in 3D reconstruction (CD & EMD). In summary ProJo4D shows consistent improvement across different material models and different object shapes across both datasets.

We also evaluate our method on the *GSO dataset* from Vid2Sim Chen et al. (2025). To ensure a fair comparison, we reran GIC and our proposed method using the same material model as Vid2Sim on the GSO dataset. Although Vid2Sim, PAC-NeRF, and GIC all adopt the Neo-Hookean material model for elastic objects, their exact formulations differ. The variant, GIC*, in Tab. 4 reports the performance of GIC with the identical material model as Vid2Sim. As the results indicate, aligning the material models leads to improved performance for GIC. Moreover, our proposed optimization strategy, ProJo4D, consistently demonstrates further performance gains.

Table 4: Future state prediction and material parameter estimation on the GSO dataset (original setting).

|  | PSNR ↑ | MAE $\log E$ ↓ | MAE $\nu$ ↓ |
|---|---|---|---|
| PAC-NeRF | 20.11 | 2.50 | 0.21 |
| Spring-Gaus | 18.32 | - | - |
| GIC | 19.20 | 2.01 | 0.16 |
| Vid2Sim | 25.07 | 0.51 | **0.06** |
| GIC* | 21.90 | 0.59 | 0.07 |
| GIC* + ProJo4D | **26.80** | **0.31** | **0.06** |

In multi-parameter inverse problems, exact recovery of every parameter is often ill-posed: multiple parameter configurations can generate nearly indistinguishable dynamics in future state simulation. Worse performance in a parameter while improvement in others does not necessarily indicate worse overall inverse physics estimation, rather the future state prediction accuracy is often a more reliable indicator of the overall estimation quality. Examples are "banana" and "paste' in Spring-Gaus dataset: while ProJo4D underperforms GIC in a single physics parameter, it consistently achieves lower CD and EMD and higher PSNR in future prediction. Taken together, ProJo4D not only improves parameter estimation overall but, more importantly, delivers reliable gains in future state predictive accuracy—an outcome that is especially relevant for downstream applications such as simulation and digital twin construction.

**Real-world Dataset.** Table 5 provides results on the Spring-Gaus real-world dataset. Since no ground truth three-dimensional mesh or material parameters are available, we evaluate using only two-dimensional metrics: PSNR and SSIM. Because Spring-Gaus works exclusively with elastic objects, we use the elastic material model for both GIC Cai et al. (2024) and our method. Consistent with the synthetic data results in Tabs. 2

Table 5: 2D future state prediction accuracies on Spring-Gaus real-world dataset Li et al. (2023b).

|  |  | bun | burger | dog | pig | potato | mean |
|---|---|---|---|---|---|---|---|
| PSNR ↑ | Spring-Gaus | 26.79 | 35.13 | 30.31 | 31.95 | 28.96 | 30.63 |
|  | GIC | 32.14 | 36.89 | 33.35 | 32.30 | 35.02 | 34.05 |
|  | GIC + ProJo4D | **37.35** | **39.01** | **36.07** | **38.90** | **40.18** | **38.30** |
| SSIM ↑ | Spring-Gaus | 0.986 | 0.995 | 0.993 | 0.994 | 0.989 | 0.991 |
|  | GIC | 0.994 | 0.995 | 0.995 | 0.996 | 0.995 | 0.995 |
|  | GIC + ProJo4D | **0.997** | **0.996** | **0.996** | **0.997** | **0.997** | **0.996** |

and 3, our method outperforms other approaches in both PSNR and SSIM on real-world images. This performance improvement demonstrates that our proposed optimization strategy significantly enhances estimation performance not only in synthetic but also in real-world settings, with additional visual results provided in the appendix.

**Ablation Study.** On the Spring-Gaus Dataset, we evaluate the robustness of different approaches under both dense and sparse view settings. As shown in Fig. 7, while Spring-Gaus and GIC exhibit significant degradation in sparse-view scenarios, resulting in drastic increases in Chamfer Distance, our method ProJo4D maintains consistently low errors. This indicates that ProJo4D effectively alleviates error accumulation under limited viewpoints, achieving superior generalization to sparse observations.

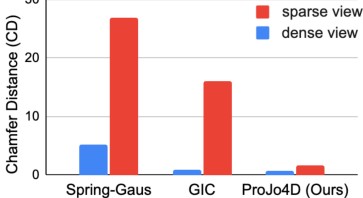

Figure 7: The average Chamfer distance of different methods under dense and sparse views.

## 5 CONCLUSION

We introduced ProJo4D, a progressive joint optimization framework that incrementally expands jointly optimized parameters guided by their sensitivity. This strategy ensures robust estimation of geometry, appearance, and physical parameters under highly ambiguous, sparse-view inputs. Evaluations on benchmark datasets show that ProJo4D consistently outperforms state-of-the-art methods in 4D future state prediction, novel view rendering, and physical parameter estimation, demonstrating practical relevance.

While ProJo4D shows strong performance in 4D scene reconstruction and inverse physics estimation from sparse-view videos, it shares limitations common to existing methods. First, it cannot overcome fundamental challenges from underlying material models, such as non-differentiable,

material-parameter-dependent branches, which require longer iterations and increase sensitivity to learning rates for some, including non-Newtonian fluids. Second, reliance on computationally intensive physics simulations, specifically MPM, results in high costs. Future work should explore accelerating simulations via neural surrogates, differentiable Neural PDEs, or other lightweight methods.

## ETHICS STATEMENT

We adhere to the ICLR Code of Ethics. No private, sensitive, or personally identifiable data are involved. The goal of this research is to advance inverse physics, and it does not raise foreseeable ethical concerns or produce harmful societal outcomes.

## REPRODUCIBILITY STATEMENT

All datasets used in our experiments are standard benchmarks that are publicly available. Upon acceptance, we will release our codebase, including training, and evaluation, along with configuration files.

## THE USE OF LARGE LANGUAGE MODELS (LLMS)

Large language models (LLMs) were employed for linguistic editing, mainly to improve readability of human written texts and correcting grammatical and spelling mistakes. The conceptualization, methodology, analysis, and core technical writing were carried out entirely by the authors.

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

## A  SUPPLEMENTARY MATERIAL

We provide additional qualitative results in the form of videos for two synthetic datasets, Spring-Gaus and PAC-NeRF, as well as the real-world dataset from Spring-Gaus. These materials are accessible via the included `index.html` file.

Table 6: Performance comparison results across different material types and configurations. We evaluate 3D future state prediction accuracy and material parameter estimation. Bold values indicate best performance for each metric and material type. X, A, S, and M denote positions, appearances, physical states, and material parameters.

|  | Stage 1 | **Elastic** | **Newtonian** | **Non-Newtonian** | **Plasticine** | **Sand** |
|---|---|---|---|---|---|---|
| CD↓ | S | $0.953 \pm 0.295$ | $6.319 \pm 7.854$ | $9.668 \pm 4.543$ | $21.891 \pm 17.848$ | $2.727 \pm 0.531$ |
|  | M | $1.020 \pm 0.314$ | $4.830 \pm 6.432$ | $9.205 \pm 3.559$ | $20.470 \pm 17.660$ | $4.067 \pm 2.469$ |
|  | SM | $1.057 \pm 0.349$ | $4.896 \pm 4.254$ | $9.682 \pm 4.126$ | $20.434 \pm 17.976$ | $2.743 \pm 0.543$ |
|  | **XAS (ProJo4D)** | $\mathbf{0.913 \pm 0.301}$ | $\mathbf{0.339 \pm 0.108}$ | $\mathbf{0.473 \pm 0.248}$ | $\mathbf{1.103 \pm 0.948}$ | $\mathbf{0.264 \pm 0.017}$ |
|  | XAM | $1.053 \pm 0.339$ | $3.226 \pm 1.607$ | $9.164 \pm 4.587$ | $20.368 \pm 16.643$ | $2.457 \pm 0.359$ |
|  | XASM (Full Joint) | $1.318 \pm 1.117$ | $0.346 \pm 0.095$ | $8.104 \pm 13.563$ | $17.678 \pm 18.170$ | $53.564 \pm 19.404$ |
| MAE $v_0$↓ | S | $\mathbf{0.007 \pm 0.003}$ | $0.074 \pm 0.045$ | $0.132 \pm 0.031$ | $0.102 \pm 0.091$ | $0.085 \pm 0.039$ |
|  | M | $0.035 \pm 0.025$ | $0.089 \pm 0.045$ | $0.154 \pm 0.052$ | $0.131 \pm 0.107$ | $0.150 \pm 0.128$ |
|  | SM | $0.013 \pm 0.009$ | $0.073 \pm 0.044$ | $0.149 \pm 0.086$ | $0.128 \pm 0.070$ | $0.095 \pm 0.034$ |
|  | **XAS (ProJo4D)** | $\mathbf{0.007 \pm 0.003}$ | $\mathbf{0.008 \pm 0.002}$ | $\mathbf{0.005 \pm 0.003}$ | $\mathbf{0.024 \pm 0.056}$ | $\mathbf{0.005 \pm 0.003}$ |
|  | XAM | $0.099 \pm 0.036$ | $0.125 \pm 0.045$ | $0.162 \pm 0.031$ | $0.173 \pm 0.091$ | $0.252 \pm 0.056$ |
|  | XASM (Full Joint) | $0.020 \pm 0.033$ | $0.008 \pm 0.004$ | $0.080 \pm 0.099$ | $0.092 \pm 0.102$ | $0.046 \pm 0.032$ |

## B  ABLATION STUDY

We investigate progressive joint optimization strategies that start by optimizing only a subset of parameters before transitioning to full joint optimization. We further compare these progressive approaches against direct full joint optimization.

To this end, we modify the parameter set optimized in Stage 1 and conduct experiments on the PAC-NeRF dataset, which contains multiple material models. Because different material models involve distinct parameterizations, we report only metrics that are common across all models. In particular, we use the Chamfer distance to evaluate the accuracy of future dynamics estimation and mean absolute error for initial velocity estimation.

As shown in Tab. 6, our progressive method demonstrates stability across different material models. Fully optimizing all parameters immediately after obtaining 3D/4D representations (XASM in Tab. 6) achieves performance comparable to our method for relatively simple material models (Elastic and Newtonian). However, it shows significant degradation for more complex material models (Non-Newtonian, Plasticine, and Sand). This suggests that while full joint optimization can perform as effectively as carefully designed optimization strategies for simpler models, it becomes unreliable as the material models become complex.

We additionally evaluate robustness with respect to the number of camera views in Tab. 7. As shown in the table, future prediction and material parameter estimation performance improve as the number of camera views increases. By adjusting only the parameter sets optimized at each stage, the performance remains robust even when the number of views decreases.

We also compare alternative optimization strategies in Tab. 8, including sequential and cyclic approaches. "Sequential" refers to the baseline, GIC. To account for the fact that joint optimization changes the effective number of iterations per parameter set, we compare ProJo4D with an extended sequential baseline ("Sequential+") that matches both the total number of iterations and the number of images used per parameter set. Table. 9 summarizes the number of iterations for each stage for the different optimization strategies. Results show that simply increasing iterations does not meaningfully improve performance. We further evaluate iterative optimization ("Cycles"), where one parameter set is optimized at a time and repeated for several cycles while keeping total iterations constant. Two configurations are tested: (XA-S-M)×4-A, which repeats Stages 0–2, and XA-(S-M-A)×4, which repeats Stages 1–3, both with 4 cycles. For example, (XA-S-M)×4-A runs Stage 0 for 10K, Stage 1 for 25, Stage 2 for 25, repeated 4 times, then Stage 3. These results indicate

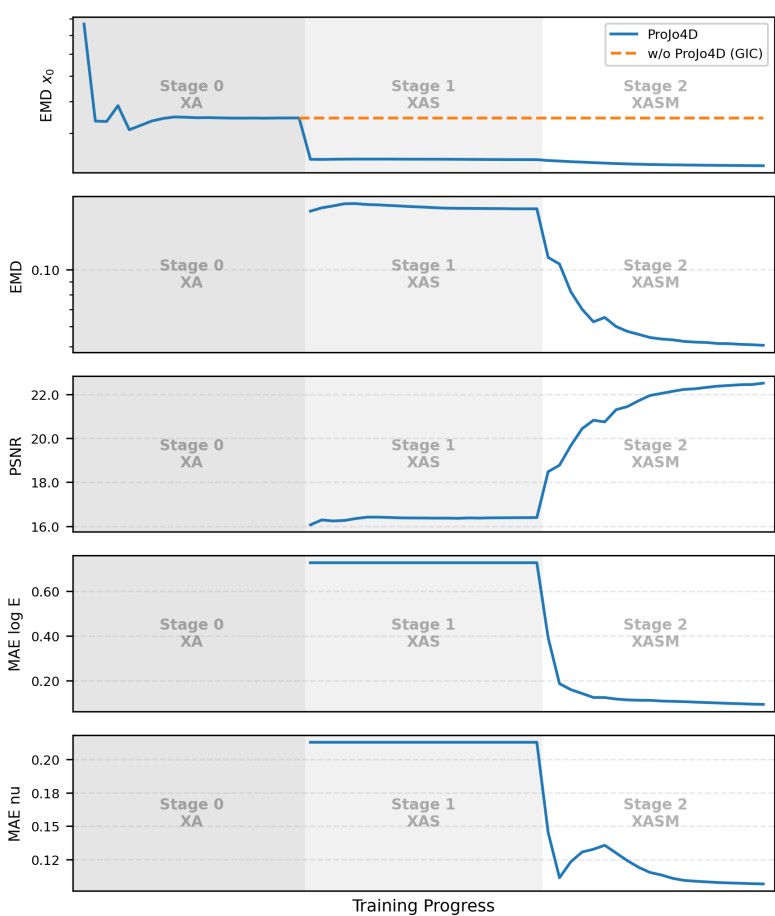

Figure 8: Average optimization trajectories for SpringGaus scenes. The left third (dark gray) corresponds to Stage 0, the middle third (light gray) to Stage 1, and the right third (white) to Stage 2. EMD and PSNR measure future-state prediction, while MAE $\log E$ and MAE $\nu$ measure material parameter estimation. EMD$x_0$ indicates the earth moving distance at time 0 (canonical space). For visualization, Stage 0 is rescaled because it has many more iterations than Stages 1 and 2. Unlike existing methods, which do not optimize positions using physics-informed gradients or refine geometry, our method introduces physics-informed gradients during optimization, resulting in better geometry even with the same visual inputs.

Table 7: Ablation study on the impact of different numbers of camera views. ProJo4D consistently improves performance across varying numbers of views.

|  |  | Number of cameras | | |
|---|---|---|---|---|
|  |  | 2 | 3 | 10 |
| CD $\downarrow$ | GIC | 12.00 | 16.11 | 0.95 |
|  | + ProJo4D | **1.66** | **1.60** | **0.65** |
| EMD $\downarrow$ | GIC | 0.129 | 0.128 | 0.049 |
|  | + ProJo4D | **0.055** | **0.057** | **0.034** |
| PSNR $\uparrow$ | GIC | 16.57 | 17.58 | 22.98 |
|  | + ProJo4D | **20.56** | **22.30** | **26.95** |
| SSIM $\uparrow$ | GIC | 0.844 | 0.850 | 0.930 |
|  | + ProJo4D | **0.880** | **0.913** | **0.951** |
| MAE $\log E \downarrow$ | GIC | 0.4951 | 0.2311 | 0.1286 |
|  | + ProJo4D | **0.1094** | **0.1043** | **0.0643** |
| MAE $\log \nu \downarrow$ | GIC | 0.2752 | 0.1790 | **0.0458** |
|  | + ProJo4D | **0.1374** | **0.0911** | 0.0654 |

Table 8: Ablation study of alternative optimization strategies. The order of optimization is represented using X (position), A (appearance), S (velocity), and M (material parameters). "Sequential" denotes standard optimization (GIC), updating one parameter set at a time. "Sequential+" runs additional iterations while keeping the same strategy, ensuring each parameter set receives at least as many iterations as in ProJo4D. "Cyclic" strategies repeat optimization over particle stages with fewer iterations per cycle, keeping the total number of iterations constant. We use 4 cycles (N=4) in our experiments. Results show that cyclic optimization improves performance over sequential strategies but does not match ProJo4D.

|  | ProJo4D
XA-XAS-XASM | Sequential
XA-S-M-A | Sequential+
XA-S-M-A | Cyclic
(XA-S-M)×4-A | Cyclic
XA-(S-M-A)×4 |
|---|---|---|---|---|---|
| CD $\downarrow$ | **1.60** | 16.11 | 16.72 | 14.63 | 3.20 |
| EMD $\downarrow$ | **0.057** | 0.128 | 0.135 | 0.136 | 0.085 |
| PSNR $\uparrow$ | **22.30** | 17.58 | 18.01 | 16.96 | 19.04 |
| SSIM $\uparrow$ | **0.913** | 0.850 | 0.854 | 0.834 | 0.882 |
| MAE $\log E \downarrow$ | **0.1043** | 0.2311 | 0.1958 | 0.2547 | 0.1616 |
| MAE $\log \nu \downarrow$ | **0.0911** | 0.1790 | 0.2984 | 0.2524 | 0.2686 |

Table 9: Iterations and image batch sizes for each stage in Tab. 8. Batch size is shown in parentheses; for stages with multiple cameras and frames, we use the format (the number of cameras × the number of frames).

|  | ProJo4D | Sequential (GIC) | Sequential+ |
|---|---|---|---|
| Stage 0 | 40K (1) | 40K (1) | 47K (1) |
| Stage 1 | 100 (3x3) | 100 (3x3) | 200 (3x3) |
| Stage 2 | 100 (3x20) | 100 (3x20) | 100 (3x20) |
| Stage 3 | 0 | 40K (1) | 40K (1) |

that introducing cyclic optimization too early, before the 4D representation is sufficiently accurate, can degrade performance. While cyclic optimization can outperform simple sequential strategies when applied after 4D representation learning, it still does not match the performance or stability of ProJo4D's progressive joint optimization.

## C  Hyperparameters

**Stage 0: 3D/4D representation learning.** To isolate the effects of physical parameter optimization from unrelated factors, we use the same 3D/4D representations as the base system. For GIC + ProJo4D, we initialize with the identical 4D representations used by GIC and run only Stage I and Stage II. For both GIC and ProJo4D, the 4D representations are obtained after 40K iterations of 4D representation learning: a 3K-iteration warmup for static scene representation followed by 37K iterations with deformation networks for full 4D modeling.

**Stage I & II.** We used the same hyperparameters as the baseline (GIC) to rule out performance gains from hyperparameter tuning. For the Spring-Gaus dataset, ProJo4D uses 100 iterations for Stage I and 100 for Stage II, matching GIC. Although GIC includes an additional 30K iterations for appearance refinement, we do not require this step because appearance is jointly optimized during Stage I and Stage II.

For the PAC-NeRF dataset, we follow the iteration schedule used in GIC for each material type:

- **Elastic**: 100 iterations for Stage I and 150 for Stage II
- **Newtonian**: 100 iterations for Stage I and 250 for Stage II
- **Non-Newtonian**: 100 iterations for Stage I and 350 for Stage II
- **Plasticine**: 100 iterations for Stage I and 300 for Stage II
- **Sand**: 100 iterations for Stage I and 150 for Stage II

## D  Experimental Details for the GSO dataset.

Although Vid2Sim, PAC-NeRF, and GIC all adopt the Neo-Hookean material model for elastic objects, their exact formulations differ. Specifically, PAC-NeRF and GIC define the Kirchhoff stress tensor $\tau$ as:

$$\tau_{PAC-NeRF} = \mu F F^T + (\lambda J - \mu) I, \tag{8}$$

where $F$ is the deformation gradient and $J$ is the determinant of $F$. Here, $\mu$ and $\lambda$ denote the Lamé parameters. In contrast, Vid2Sim follows the formulation used in Simplicits, where the Kirchhoff stress tensor is defined as:

$$\tau_{Simplicits} = \mu F F^T + (\lambda(J-1) - \mu) J I. \tag{9}$$

To ensure a fair comparison, we reran GIC and our proposed method using the same material model (Eq. 9) on the GSO dataset.

