# OpenReview forum: "ProJo4D: Progressive Joint Optimization for Sparse-View Inverse Physics Estimation"
_ICLR.cc/2026/Conference — Submitted to ICLR 2026_

### Official Review · Reviewer_gXFz · 2025-10-19

**Soundness:** 3
**Presentation:** 3
**Contribution:** 2
**Rating:** 6
**Confidence:** 2

**Summary:**

This paper proposes ProJo4D, a progressive joint optimization framework for inverse physics estimation from sparse-view video. It identifies a key failure mode in prior sequential optimization methods: error accumulation from poor initial 3D reconstruction under sparse-views. ProJo4D avoids this by using a 3-stage curriculum: it first learns a 4D representation, then jointly optimizes for initial physical states (e.g., velocity), and finally conducts a full joint optimization including material properties. This progressive strategy is shown to dramatically outperform state-of-the-art methods like GIC and Spring-Gaus on standard sparse-view benchmarks.

**Strengths:**

The paper addresses a critical and practical bottleneck—the failure of physics estimation from sparse views. This is highly relevant for real-world robotics and XR applications.

The experimental validation is exceptionally strong. The method demonstrates not just marginal but order-of-magnitude improvements (e.g., CD 16.11 $\rightarrow$ 1.60) over strong baselines on multiple datasets.

**Weaknesses:**

The abstract claims the optimization order is "guided by their sensitivity", but this concept is never defined, quantified, or justified in the paper. The chosen order (velocity, then material) appears to be an empirically-driven heuristic, not a formal principle.

The paper convincingly shows that ProJo4D works, but not how it avoids error accumulation. It lacks a crucial analysis (directly comparing the optimization trajectory of ProJo4D against the sequential (GIC) and full-joint (XASM ) methods.

**Questions:**

Could you please clarify the "sensitivity"  claim? Is the optimization order based on a formal analysis, or physical intuition?

Given the strong performance of the "XASM" ablation on simpler materials, would you agree that a more precise claim is that your progressive strategy acts as a robust regularizer for joint optimization, making it viable for complex materials where a naive joint approach fails?

---

> ### Author Response · Authors · 2025-11-18
>
> Dear Reviewer gXFz,
>
> We thank you for your sharp observations and constructive feedback. Your questions have helped us significantly improve the clarity and framing of our core contributions. We have uploaded a revised manuscript and supplementary material that directly addresses your points.
>
> Below, we provide a point-by-point response.
>
> **Clarification of Sensitivity (Q1, W1)**
> “Sensitivity” unintentionally suggested a notion that is not part of our method, and we removed it to avoid confusion. Our optimization order follows the empirical reasoning widely adopted in prior inverse-physics work (e.g., PAC-NeRF, SpringGauss, GIC, MASIV), where early stages focus on quantities most directly tied to initial dynamics.
> As shown in Figs. 3 and 4, under sparse views, sequential optimization tends to accumulate errors, while full joint optimization often becomes unstable. These observations motivated us to delve into optimization strategies.
> To further support the choice of ordering, Tab. 6 presents a direct comparison between velocity-first (XAS) and material-first (XAM) variants. XAS consistently achieves higher accuracy across all settings, demonstrating that our ordering is empirically well-supported.
>
> **Progressive Strategy as a Robust Regularizer (Q2)**
> Yes, this interpretation aligns with our findings. The full-joint baseline (XASM) performs similarly to our method for simple elastic materials (Tabs. 3 and 6). However, as material models become more nonlinear or contain non-differentiable branches, joint optimization becomes highly unstable, often collapsing into poor local minima. Our progressive strategy acts as a robust regularizer, preventing these catastrophic situations and enabling stable optimization while addressing accumulated errors.
>
> **Error Accumulation Mechanism (W2)**
> ProJo4D avoids error accumulation by allowing physics-informed gradients to jointly update geometry and physical parameters throughout the optimization. This contrasts with sequential pipelines (GIC), where early geometry errors are “locked in” and cannot be corrected later. We provide a performance-over-time analysis in Fig. 8 (Supplementary), showing that ProJo4D continues to improve 3D shape estimation accuracy (EMD x_0) in later stages, supporting our claim that early errors can be corrected.
> Fig. 4 visualizes the optimization trajectories of all three strategies: Sequential (GIC), Joint, and Progressive (ProJo4D). Tab. 3 has been updated to explicitly separate these strategies across material models, making the contrast between error accumulation (Sequential), instability (Joint), and stable convergence (Progressive) clearer.
> A compact version of Tab. 3 is provided here; please refer to the main paper for the full table.
>
> | Metric | Method | Elasticity | Newtonian | Non-Newtonian | Plasticine | Sand |
> |--------|---------|------------|-----------|---------------|------------|--------|
> | CD | GIC | 5.512 | 0.537 | 0.689 | 2.012 | 20.262 |
> | CD | +ProJo4D | 0.913 | 0.339 | 0.473 | 1.103 | 0.264 |
> | CD | +Full Joint | 1.318 | 0.346 | 8.104 | 17.678 | 53.564 |
> | MAE v_0 | GIC | 0.008 | 0.009 | 0.015 | 0.010 | 0.007 |
> | MAE v_0 | +ProJo4D | 0.007 | 0.008 | 0.005 | 0.024 | 0.005 |
> | MAE v_0 | +Full Joint | 0.020 | 0.008 | 0.080 | 0.092 | 0.046 |

---

> > ### Author Response · Authors · 2025-11-25
> >
> > Dear reviewer gXFz,
> >
> > Thank you again for your thoughtful and detailed feedback. If you have any further questions or concerns regarding our rebuttal or the paper, we'd be happy to clarify or provide additional context.

---

### Official Review · Reviewer_mKC6 · 2025-10-30

**Soundness:** 2
**Presentation:** 3
**Contribution:** 2
**Rating:** 4
**Confidence:** 4

**Summary:**

The proposed ProJo4D framework aims to jointly estimate 3D Gaussian parameters, deformation models, initial velocity, and material properties in a progressive manner. Specifically, ProJo4D highlights its effectiveness under a sparse-view setting, where it outperforms other baselines in terms of future state prediction and physical parameter estimation.

**Strengths:**

1. The presentation is clear, and the background is investigated comprehensively.

2. The optimization design of ProJo4D framework is neat overall.

**Weaknesses:**

1. The Deformation Network is only optimized at stage 0, which may not be correct. The later estimation for initial velocity and material parameters through physics simulation will rely on the predicted positions from the Deformation Network, which may not be reliable.

2. It is unclear what the key is to the optimization strategy of ProJo4D. Is it the progressive estimation for different kinds of parameters? Or is it due to some parameters are optimized repeatedly across multiple stages? More ablations are needed.

3. Experiments can be fairer and more comprehensive:

(1) It is suggested to compare the optimization cost (e.g., how many rounds of optimization are needed to estimate some parameters).

(2) It is suggested to provide comparison results for dense views as well, and provide analysis on why ProJo4D is less sensitive to sparse view.

**Questions:**

1. For the comparison between different optimization strategies, can you make the total number of optimization stages comparable? For example, according to Table 1, position (X) in ProJo4D is optimized 4 times, while it is only optimized once in GIC. Experiments should be adapted accordingly to be fair.

2. More ablations on the optimization strategy are needed. For example, instead of progressive optimization, what if parameters are estimated sequentially (X,A->S->M) but optimized for multiple rounds?

3. Is it necessary to include the Diff. Physics Simulation branch, given that the future state prediction is the ultimate goal as mentioned in line 442? Without the Physics Simulation branch, can you still estimate a reasonable future states from 3D Gaussian parameters and deformation models?

---

> ### Author Response · Authors · 2025-11-18
>
> Dear Reviewer mKC6,
>
> We sincerely thank you for your detailed and insightful questions. Your feedback prompted us to run several new, targeted ablations that we believe have significantly clarified our core contribution and strengthened the paper. We have uploaded a revised manuscript and supplementary material with these new results.
>
> **Fair Comparison of Optimization Iterations (Q1, W2, W3.1)**
> To ensure a fair comparison, we added a new ablation (Tab. 8, Supplementary) that matches the total number of optimization iterations per parameter set (Sequential+).
> Please refer to Tab. 9 in the supplementary for the detailed number of iterations for each stage.
> The table below (please refer to Tab. 8 for the full version) shows that longer iterations alone do not resolve the fundamental challenges in sparse-view settings. This demonstrates that how the parameters are optimized is the critical factor, rather than simply the total number of iterations.
>
> | Metric | ProJo4D | Sequential (GIC) | Sequential+ |
> |--------|---------|------------------|-------------|
> | CD | 1.60 | 16.11 | 16.72 |
> | PSNR | 22.30 | 17.58 | 18.01 |
> | MAE log E | 0.1043 | 0.2311 | 0.1958 |
> | MAE log nu | 0.0911 | 0.1790 | 0.2984 |
>
>
> **Comparison with Cyclic Optimization (Q2)**
> We also evaluated cyclic optimization strategies with multiple rounds, keeping the total number of iterations fixed. Two variants were tested:
> (XA-S-M)×4-A: Stage 0–2 cycles followed by Stage 3
> XA-(S-M-A)×4: Stage 0 followed by a cycle of Stages 1–3
> Here, X denotes positions, A appearance, S velocity, and M material parameters. For both variants, we used 4 cycles and reduced the number of iterations per stage to maintain the total iteration budget. For example, (XA-S-M)×4-A runs Stage 0 for 10K, Stage 1 for 25, Stage 2 for 25, repeated 4 times, then Stage 3.
> Although cyclic optimization improves over standard sequential optimization, it still cannot match the performance of our progressive joint optimization. This confirms that the benefit arises from the structure of progressive joint optimization rather than simply additional iterations.
> The full table is in Tab. 8.
>
> | Method | CD ↓ | PSNR ↑ | MAE log E ↓ |
> |--------|------|--------|-------------|
> | ProJo4D | 1.60 | 22.30 | 0.1043 |
> | Cyclic (XA-S-M)×4-A | 14.63 | 16.96 | 0.2547 |
> | Cyclic XA-(S-M-A)×4 | 3.20 | 19.04 | 0.1616 |
>
> **Role of the Physics Simulation Branch (Q3)**
> The Stage-0 deformation model in both GIC and ProJo4D is optimized to reconstruct the 4D representation over the training video and serves only as 3D guidance during optimization. By itself, it cannot support future-state prediction, as it lacks physical consistency. The differentiable physics simulator is essential to accurately estimate material parameters and predict physically plausible dynamics. Moreover, ProJo4D leverages this simulator to further improve 3D geometry accuracy (EMD x_0 in Fig. 8), extracting information beyond what the deformation model alone can provide.
>
> **Potential Errors from Deformation Network (W1)**
> This is precisely the problem our method addresses. In sparse-view settings, 3D/4D representations are inherently noisy due to limited visual information, with or without a deformation network. Our optimization strategy leverages gradients from a differentiable physics simulator to jointly refine geometry and physical parameters in later stages, remaining robust not only to noisy initial estimates but also to imperfect guidance during optimization.
>
> **Robustness to Different Number of Views (W3.2)**
> We do have results for this in Fig. 7 of the main paper, which compares GIC and GIC + ProJo4D in both sparse and dense settings. To make this more comprehensive, we have also added experiments across varying numbers of camera views (Table 7 in the Supplementary). The results confirm that ProJo4D consistently improves performance in all scenarios (sparse and dense), though the benefit is most dramatic in the challenging sparse-view settings (e.g., 86% reduction in CD and 78% reduction in MAE log E under 2-camera settings).
> Based on the analysis regarding error accumulation, **we believe ProJo4D is less sensitive to sparse views as it can obtain information from differentiable physics simulators without requiring additional visual inputs and alleviate error accumulation, while being much more robust and stable than full joint optimization.**
>
> | Metric | Method | 2 cameras | 3 cameras | 10 cameras |
> |--------|---------|-----------|-----------|------------|
> | CD | GIC | 12.00 | 16.11 | 0.95 |
> | CD | +ProJo4D | 1.66 | 1.60 | 0.65 |
> | PSNR | GIC | 16.57 | 17.58 | 22.98 |
> | PSNR | +ProJo4D | 20.56 | 22.30 | 26.95 |
> | MAE log E | GIC | 0.4951 | 0.2311 | 0.1286 |
> | MAE log E | +ProJo4D | 0.1094 | 0.1043 | 0.0643 |
>
> If you have any remaining concerns, we hope to address them during the discussion period.

---

> > ### Author Response · Authors · 2025-11-25
> >
> > Dear reviewer mKC6,
> >
> > Thank you again for your thoughtful and detailed feedback. If you have any further questions or concerns regarding our rebuttal or the paper, we'd be happy to clarify or provide additional context.

---

> > ### Comment · Reviewer_mKC6 · 2025-11-26
> > **Comments**
> >
> > Thank you for your efforts in addressing the reviewer's concerns.
> >
> > The additional experiments on various optimization strategies have made the current comparisons more comprehensive and fair.
> >
> > However, while the proposed optimization strategy demonstrates effectiveness and yields significant performance gains on sparse-view GIC, two weaknesses remain:
> >
> > 1) Generalization Across Methods and Data: The general applicability of this optimization strategy to different methods and datasets is unclear. Simply showcasing its effectiveness on a single baseline (GIC) weakens the contribution.
> >
> > 2) Lack of In-Depth Analysis: Despite improved experimental results, there is a lack of thorough analysis. For instance, the question "Why is ProJo4D less sensitive to sparse views?" is not convincingly addressed. It is important to note that “information from differentiable physics simulators” is also accessible to GIC, and cyclic optimization can provide stability as well.
> >
> > Overall, the concept is innovative, and the performance is promising. However, demonstrating the generalization ability of the optimization strategy and providing in-depth explanations for the observed performance gains, whether through analytical or experimental means, would significantly enhance the paper's robustness and completeness for future submission.

---

### Official Review · Reviewer_hfpm · 2025-10-31

**Soundness:** 2
**Presentation:** 3
**Contribution:** 2
**Rating:** 2
**Confidence:** 4

**Summary:**

This paper proposes ProJo4D, a progressive joint optimization framework to learn physics parameters from sparse-view videos. In general, three stages are included in the training pipeline, progressively from learning a coarse 3D/4D representation to learning full set of physical parameters. Obvious performance gains are shown in both synthetic and real-world dataset.

**Strengths:**

1. The paper expresses its main target very clearly, making it easy to follow
2. Although the method is simple, the improvement is obvious.

**Weaknesses:**

Although the method shows great performance in the experiments, there are still some conceptual or experimental weaknesses:
1. Since the stage 0 is purely learned with RGB supervision with no other constraints, it is possible that the learned deformation field does not obey the physical smoothness. Once the deformation field is learned, it will not be optimized further. So if there is issue in the first stage, the issue will accumulate to the later stages. The influence of enabling further refinement should be analyzed in later stages.
2. The implementation details for the 4D representation learning stage is not well-elaborated.
3. Ablation is not thorough enough. Performances after different stages should be reported.
4. Whether the performance gain is from the different ways in learning 4D representation or from the whole pipeline is not clear. The authors should report the reconstruction performance comparison between stage 0 and the 4D rep of GIC.
5. It seems the ability to enble better sparse-view reconstruction is from the stage 0, which is not new. And the parameter estimation pipeline is not new either. So I doubt the novelty of this work.

In general, where the performance gain comes from is not clearly demonstrated by the experiments in my opinion. So I can’t support the paper to be accepted. But I am willing to modify my ratings if the authors can address my concerns.

**Questions:**

1. Do ProJo4D and GIC share the same 4D reconstruction framework at the first stage?
2. Please refer to weaknesses.

---

> ### Author Response · Authors · 2025-11-18
>
> Dear Reviewer hfpm,
>
> Thank you for your detailed and constructive feedback. We greatly appreciate your careful reading of our work and the opportunity to clarify the contributions and performance improvements of ProJo4D. Below, we provide a detailed response to each of your points, emphasizing how our method achieves its gains and addressing your concerns regarding experimental evidence.
>
> **Shared 4D Reconstruction Framework (Q1, W2, W4)**
> We would like to clarify that **ProJo4D uses the exact same 3D/4D and physics representations of GIC** and only focuses on improving optimization strategies, which lead to significant performance improvement. We do not modify the 4D representation learning from Stage 0 in any way and start from the same representation; hence, both GIC & ProJo4D have the exact same result at the end of Stage 0, and the only differences arise in later optimization stages. We have added more implementation details in the Supplementary (Sec. C) to make this clear.
>
> **Potential Errors from Stage 0 (W1)**
> Your concern about noisy Stage-0 representations is exactly the problem we tackle.
> Noisy Stage-0 4D representations pose a fundamental challenge in sparse-view settings, affecting both initialization and 3D guidance. ProJo4D addresses this by leveraging physics-informed gradients from a differentiable simulator to jointly and stably refine geometry and physical parameters. As shown in Fig. 8 (EMD x_0) and Tables 3 & 6 (XASM), longer Stage-0 optimization saturates (EMD x_0 in Fig. 8 and Sequential+ in Tab. 8), and naive joint optimization is unstable (Updated Tab.3 and XASM in Tab. 6), whereas our method consistently improves shape recovery from the same inputs.
>
> **Ablation Over Different Stages (W3)**
> We have updated Figure 8 in the supplementary material to illustrate how performance evolves over time. The main insight, now highlighted in this figure, is that ProJo4D leverages physics-informed gradients from later stages to improve 3D shape reconstruction accuracy in the canonical space. This mechanism helps our method reduce the error accumulation observed in Stage 0. In contrast, the baseline’s sequential optimization is unable to correct these geometric errors, resulting in stagnant performance.
>
> **Novelty of the Work (W5)**
> A key contribution of our work is to identify optimization strategy as a fundamental problem in inverse physics estimation, which is especially important in sparse-view settings, and propose a simple yet robust solution to tackle it.
> Optimization is critical in inverse physics estimation due to the highly non-convex nature of the problem. While prior works such as PAC-NeRF, Spring-Gaus, GIC, Vid2Sim, and MASIV have advanced scene representation and physical modeling, they have largely underemphasized effective optimization, a gap ProJo4D addresses.
> ProJo4D introduces a progressive joint optimization strategy that is both intuitively motivated and empirically validated. Its multi-stage design mitigates the error accumulation seen in prior pipelines. Though conceptually straightforward, it delivers substantial gains, highlighting optimization as a key yet underexplored component of inverse physics.
> In addition, **we believe ProJo4D is less sensitive to sparse views as it can obtain information from differentiable physics simulators without requiring additional visual inputs and alleviating error accumulation, while being much more robust and stable than full joint optimization**.
>
>
> We hope this clarifies the source of ProJo4D’s performance improvements. We would be happy to address any further questions or concerns during the rebuttal.

---

> > ### Comment · Reviewer_hfpm · 2025-11-19
> >
> > I'll first appreciate the authors' efforts in trying to address most of the concerns of me and of other reviewers.
> >
> > After reading the reviews from other reviewers and the responses from the authors, especially the new added ablations in the supp, I believe most of my concerns are addressed so I am willing to increase my recommendation. However, I cannot increase the rating too much at the current stage based on the following emerging concerns, so I decide to hear from the authors further before I make my final decision.
> >
> > 1. **Novelty**: the main contribution of this work is an optimization framework by redesigning the order and sequence of parameters to be learned, with no other things new. It's efficient but simple. I'm not saying a good contribution must be complex, but the overall contribution is somewhat incremental. This hinders me from strongly advocate this paper.
> > 2. **Limited demonstration**: Although the scheme of the introduced optimization framework is clear now, but the generalizability of the framework is still not demonstrated. As the selling point of the paper is a framework but not a model, the authors should show it on diverse backbone models. I understand the tight time so at least one more backbone models should be shown, such as Vid2sim or Spring-Gaus or MASIV,  where similar performance gains should be shown to demonstrate this is not an incremental contribution only to GIC.
> >
> > I tried to propose the new request for the new experiments as early as possible so the authors can have enough time. I understand the tight timeline, so I will wait for the results till the last minute.

---

> > > ### Author Response · Authors · 2025-11-25
> > >
> > > We sincerely appreciate your constructive feedback and your willingness to reconsider the score. We have taken your advice regarding generalization seriously and have performed the requested additional experiments.
> > >
> > > **Generalization: ProJo4D on Other Methods**
> > > As requested, to demonstrate whether ProJo4D can be applicable to other methods, we tested it on a different backbone: Spring-Gaus.Spring-Gaus differs from GIC in several core design choices (e.g., the use of anchors rather than Gaussians for physical simulation, different optimization steps, and losses). We integrated the ProJo4D strategy by modifying only the velocity and material parameter estimation phases:
> > > - Velocity Phase: Adjusted to jointly optimize velocity, positions, and appearance.
> > > - Material Phase: Adjusted to jointly optimize all physical parameters.
> > >
> > > Because Spring-Gaus updates geometry and physics differently, some changes were necessary to ensure proper gradients and stable training. With these adjustments, 4D Future Prediction measured with Chamfer Distance improved from 24.54 → 11.89. However, if we apply the same structural modifications without progressive optimization to SpringGaus, its 4D future prediction performance improves from 24.54 -> 12.83. Although the gains are smaller than those observed with GIC, this is consistent with the constraints of Spring-Gaus: the use of anchors with fixed connections, reduced 3D spatial flexibility, and the absence of 3D perceptual guidance limit the benefits that ProJo4D can provide.
> > >
> > > We also examined additional methods. Vid2Sim builds on Simplicits, which does not natively support initial velocity and only estimates material parameters, making a direct integration infeasible. We have also tried integrating with MASIV, but the training time is slow, and hence any future integration will require more time. We note these limitations transparently to avoid overstating generality.
> > >
> > > In summary, our method achieves state-of-the-art performance, indicating that strong baselines combined with a well-structured progressive optimization strategy are key to reliable future prediction and physical parameter estimation from sparse views.

---

> > > > ### Comment · Reviewer_hfpm · 2025-11-25
> > > >
> > > > Thanks for the authors' efforts in taking new experiments promptly. I appreciate it a lot. Based on the author's response and the results from the new experiment, I decide to increase my score to **4** with the following reasons:
> > > >
> > > > 1. The framework is simple and general to some extend, which can benefit other physical parameter estimation models. And the authors have demonstrated this point in the new experiment. Furthermore, the new ablation studies in the response to other reviewers fully elaborate the ability of the framework, which leads to a complete paper with solid experiment. So if the paper is accepted, it's totally acceptable for me.
> > > > 2. Existing weaknesses hinder me from giving higher score:
> > > >
> > > > 	a. **Novelty:** as listed above, the main contribution of this work is to choose a right optimization sequence in learning physical parameters, which is more like a technical trick to me. Therefore, I think the novelty is subtle and incremental.
> > > >
> > > > 	b. **Generalization:** Although the authors show the model can be adapted to other backbones such as Spring-Gaus, this experiment also reminds me one more thing: this pipeline can only benefit models or pipelines with similar designs with GIC and PAC-NeRF and Spring-Gaus, *i.e.* geometry -> initial velocity -> physical parameters. For some other models in this field such as models with implicit constitutive laws, this framework is not that so instructive. In general, I think the contribution of the paper is limited to the community.
> > > >
> > > > Overall, this is a complete paper with a simple but efficient idea, but the novelty and contribution is limited. Therefore, I decide to finally recommend a **4** score. But I don't mind if the paper is accepted, if other reviewers and ACs think the contribution is not as incremental as I think.

---

### Official Review · Reviewer_9div · 2025-10-31

**Soundness:** 3
**Presentation:** 3
**Contribution:** 2
**Rating:** 4
**Confidence:** 3

**Summary:**

ProJo4D innovatively proposes a "progressive joint optimization" paradigm, which addresses the core challenges of "error accumulation" and "non-convex optimization" in inverse physics estimation under sparse-view settings. Its "visual-physical bidirectional constraint" design provides a new perspective for the integration of neural rendering and physical simulation. The paper presents solid experimental results covering diverse scenarios and material types, and holds significant reference value for downstream applications such as robotic digital twins and XR physical interactions.

**Strengths:**

1.  Innovative Progressive Joint Optimization Paradigm Solves Core Sparse-View Challenges. ProJo4D addresses the two critical bottlenecks of inverse physics estimation under sparse views: error accumulation in sequential optimization and trapping in local minima in full joint optimization, by proposing a stage-wise variable expansion strategy.
2. End-to-End Integration of 4D Dynamic Representation and Differentiable Physics Simulation. The framework tightly couples 3D Gaussian Splatting-based 4D dynamic representation with Material Point Method (MPM)-based differentiable physics simulation, forming bidirectional constraints between visual observation and physical laws.
3. Comprehensive and Rigorous Experimental Validation. Experiments cover diverse scenarios, material types, and view settings, ensuring convincing results.
4. Robustness to Complex Materials and Sparse Views. ProJo4D maintains stable performance across diverse material models.

**Weaknesses:**

1. The paper does not report key efficiency indicators such as per-frame optimization time or GPU memory usage.
2. It requires manual designation of material types and cannot handle unknown or mixed materials, limiting applicability to real-world scenes.
3. Experiments focus on 3-view settings, with no results for 2-view or single-view scenarios (common in real-world robotic monocular observation). It is unclear whether ProJo4D can maintain accuracy when view count drops further.
4. The "low-sensitivity first" optimization order is based on qualitative observation rather than quantitative analysis.
5. The weights of rendering loss and geometric loss are set empirically, with no analysis of how weights affect performance across materials/scenes.
6. The compared methods are relatively outdated, making it impossible to demonstrate the breakthrough of ProJo4D.

**Questions:**

1. The paper does not report key efficiency indicators such as per-frame optimization time or GPU memory usage.
2. It requires manual designation of material types and cannot handle unknown or mixed materials, limiting applicability to real-world scenes.
3. Experiments focus on 3-view settings, with no results for 2-view or single-view scenarios (common in real-world robotic monocular observation). It is unclear whether ProJo4D can maintain accuracy when view count drops further.
4. The "low-sensitivity first" optimization order is based on qualitative observation rather than quantitative analysis.
5. The weights of rendering loss and geometric loss are set empirically, with no analysis of how weights affect performance across materials/scenes.
6. The compared methods are relatively outdated, making it impossible to demonstrate the breakthrough of ProJo4D.

---

> ### Author Response · Authors · 2025-11-18
>
> Dear Reviewer 9div,
>
> Thank you for the time and constructive feedback. We appreciate your careful evaluation, and we have updated both the paper and the supplementary material accordingly. Major revisions are highlighted in blue. Below, we address each weakness in order.
>
> **Efficiency Indicators (W1)**
> We modify only the set of parameters optimized at each step, without introducing new modules or additional computation. Consequently, both optimization time and GPU memory usage remain effectively identical to the baseline in practice.
> For example, in the apple scene, the baseline GIC takes 10.90 s/iteration, while GIC + ProJo4D takes 11.01 s/iteration, a negligible difference well within typical variance. This confirms that our method does not incur practical efficiency overhead.
>
> **Requirement for Manual Material Type Specification (W2)**
> Our work follows previous inverse physics estimation models like (GIC, PAC-NeRF, Vid2Sim, etc.), which also assumes known material models. Material-agnostic approaches, which replace predefined constitutive models (e.g., Neo-Hookean or Drucker–Prager) with general-purpose neural material models, are promising. But the most recent paper in this direction, MASIV [ICCV’25], shows performance comparable to GIC on future state prediction, whereas ProJo4D surpasses GIC by a significant margin.
> The goal of our method is to investigate optimization strategies in inverse physics estimation, which are orthogonal to whether a predefined or neural material model is used. In theory, it is perfectly plausible to perform a progressive joint optimization strategy for a material-agnostic pipeline, as they are complementary.
>
>
> **Limited View Settings (W3)**
> We include new experiments comparing the performance of GIC and ProJo4D for 2, 3, and 10 views. Full results are now included in Table 7 of the Supplementary Material. These new results confirm that ProJo4D consistently improves performance across all metrics and view counts, demonstrating its robustness beyond the initial 3-view setting.
> We note that existing 4D representation learning pipelines are not designed for monocular settings, often producing extremely inaccurate geometry (e.g., overly large reconstructions), which prevents simulation even for the baseline. Thus, we include 2-view as the lowest viable setting.
>
> | Metric | Method | 2 cameras | 3 cameras | 10 cameras |
> |-|-|-|-|-|
> |CD|GIC|12.00|16.11|0.95|
> |CD|+ProJo4D | 1.66 | 1.60 | 0.65|
> | PSNR | GIC | 16.57 | 17.58 | 22.98|
> | PSNR | +ProJo4D | 20.56 | 22.30 | 26.95|
> | MAE log E | GIC | 0.4951 | 0.2311 | 0.1286|
> |MAE log E|+ProJo4D|0.1094|0.1043|0.0643|
>
> **Evidence for the Optimization Ordering (W4)**
> Our strategy follows a qualitative observation, consistent with strategies used in prior work (PAC-NeRF, SpringGaus, GIC, MASIV).​​ Beyond this intuition, we provide quantitative evidence: optimizing velocities first (XAS) consistently yields better accuracy than optimizing material models first (XAM), as shown in Tab. 6. This demonstrates that the ordering is not only intuitive but also empirically validated across different material models.
>
> **Sensitivity Analysis of Loss Weights (W5)**
> To ensure fairness, we kept all hyperparameters, including loss weights, identical to GIC, so improvements arise solely from optimization strategy, ProJo4D.
> To further address this, we have included a new sensitivity analysis on the loss weights. As shown in the table below, ProJo4D consistently and significantly outperforms the baseline across different weightings. This demonstrates that while further tuning could potentially improve both methods, our strategy provides a robust improvement regardless of the specific weights.
>
> |Metric|Limg=0.5, Lgeo=1.5||Limg=1, Lgeo=1 (default)||Limg=1.5, Lgeo=0.5||
> |-|-|-|-|-|-|-|
> ||GIC|+ProJo4D|GIC|+ProJo4D|GIC|+ProJo4D|
> |CD|15.63|1.42|16.11|1.60|15.62|1.56|
> |EMD|0.14|0.05|0.13|0.06|0.13|0.06|
> |PSNR|17.60|21.98|17.58|22.30|17.86|22.23|
> |SSIM|0.85|0.91|0.85|0.91|0.86|0.91|
> |MAE log E|0.25|0.11|0.23|0.10|0.17|0.11|
> |MAE nu|0.22|0.10|0.18|0.09|0.32|0.11|
>
> **“Outdated” Baselines (W6)**
> Our main baselines include Vid2Sim (CVPR 2025), GIC (NeurIPS 2024), Spring-Gaus (ECCV 2024), and PAC-NeRF (ICLR 2023), which were all SOTA or highly relevant at the time of submission. A more recent work, MASIV (ICCV 2025), was published and code-released after the ICLR submission deadline, making a direct comparison infeasible. MASIV shows performance comparable to GIC on future state prediction, whereas ProJo4D surpasses GIC by a significant margin. Furthermore, we note that MASIV's contribution (integration with a neural material model) is orthogonal to our contribution (an optimization strategy). In case we missed any other state-of-the-art method with and without publicly available code, please do let us know. We would try our best to make a reasonable comparison.
>
> We hope these address your concerns and thank again for helping us improve the paper.

---

> > ### Author Response · Authors · 2025-11-25
> >
> > Dear reviewer 9div,
> >
> > Thank you again for your thoughtful and detailed feedback. If you have any further questions or concerns regarding our rebuttal or the paper, we'd be happy to clarify or provide additional context.

---

### Author Response · Authors · 2025-12-01

Dear Area Chair and Reviewers,

We sincerely thank the reviewers for their constructive feedback and engagement. Below, we summarize the critical experimental updates added during the discussion period, which address the primary concerns regarding robustness, comparison with alternatives, and generalization.

1. **Robustness Across Varying Numbers of Cameras**

To address concerns regarding the number of views, we expanded our evaluation to include 2-view, 3-view, and 10-view scenarios.
- **Result**: ProJo4D consistently outperforms the baseline in all settings. Most significantly, in the challenging 2-view setting, ProJo4D reduces Chamfer Distance (CD) by $\approx 86\%$ ($12.00 \rightarrow 1.66$) and Material Parameter Error (MAE $\log E$) by $\approx 78\%$ ($0.495 \rightarrow 0.109$) compared to GIC.
- **Takeaway**: This confirms that ProJo4D is highly robust across varying camera setups.

2. **Comparison with Alternatives**

To strictly isolate the source of our performance gains, we introduced two distinct baselines to control for compute and strategy respectively:
- Sequential+ (Compute Control): Extends the baseline training to match the effective parameter update counts of ProJo4D, accounting for parameters that receive more updates during joint optimization.
- Cyclic (Strategy Control): Adopts an alternating optimization scheme. This tests against the most logical alternative optimization design.
- **Result**: Both baselines significantly underperform compared to ProJo4D. Sequential+ yielded a CD of 16.72 and Cyclic a CD of 3.20, whereas ProJo4D achieved a CD of 1.60.
- **Takeaway**: This confirms that our results stem specifically from the proposed progressive joint optimization.

3. **Generalization to Other Backbones**

To demonstrate that ProJo4D is a generalizable framework rather than a model-specific improvement, we integrated our optimization strategy into Spring-Gaus.
- **Result**: ProJo4D improved Spring-Gaus’s future state prediction, reducing CD from $24.54 \rightarrow 11.89$.
- **Takeaway**: This demonstrates that ProJo4D can be applied to other existing methods.

4. **Significance and Contribution**

As visualized in our updated analysis (Fig. 8, Tab. 3), ProJo4D enables physics gradients to retrospectively correct initial geometry errors (Stage 0 artifacts) while being much more robust and stable than full joint optimization.

We have updated the manuscript and supplementary material with these results.

Thank you for your time and final consideration.

---

### Meta-Review · Area_Chair_5bsd · 2026-01-02

**Summary:**

In this paper, the authors propose ProJo4D, a progressive joint optimization schedule for sparse-view inverse physics estimation, as an alternative to purely sequential or fully joint optimization. The method keeps the underlying 4D representation and differentiable physics simulator (as in GIC) unchanged, and instead modifies how and when variables are jointly optimized across stages.

While the rebuttal addresses some of the raised concerns, several key issues remain: 1) the novelty appears incremental, largely centered on an optimization schedule; 2) the generality beyond GIC-style pipelines is still not convincingly demonstrated; and 3) the paper lacks a clear, in-depth explanation of why this progressive strategy consistently works better, especially under sparse-view settings. Given that the overall reviewer sentiment remains largely borderline, I recommend rejection.

**Reviewer Concerns:**

In the initial reviews, reviewers raised concerns about 1) the fairness of the comparisons (e.g., whether gains come from more updates/iterations rather than the proposed schedule), 2) the fact that stage 0 is learned primarily from RGB supervision and may not be physically consistent, 3) incremental novelty (largely an optimization schedule rather than a new model), 4) unclear generality beyond GIC-like pipelines, and 5) a lack of in-depth explanation/analysis for why the progressive strategy is more robust under sparse views. During rebuttal, the authors provided additional ablations and experiments that address some of these points (especially fairness and robustness), but several concerns remained open, as mentioned above.

**Reviewer Scores:**

One reviewer changed the score from 2 to 4, while the others keep theirs the same. All the reviewers seems borderline with the paper.

---

### Decision · Program_Chairs · 2026-01-26

Reject